# Health-state utility of patients with HER2-positive breast cancer in Vietnam: A multicenter cross-sectional study

Tram Nguyen Nguyet Luu[1], Dai Xuan Dinh[2]*, Thinh Xuan Tran[3], Thang Binh Tran[4], Huong Thanh Tran[5,6], Kiet Huy Tuan Pham[7], Huong Thi Thanh Nguyen[2]*

**1** Faculty of Pharmacy, Hue University of Medicine and Pharmacy, Hue University, Hue, Vietnam, **2** Faculty of Pharmaceutical Management and Economics, Hanoi University of Pharmacy, Hanoi, Vietnam, **3** Department of Anesthesiology and Intensive Care, Hue University of Medicine and Pharmacy, Hue University, Hue, Vietnam, **4** Faculty of Public Health, Hue University of Medicine and Pharmacy, Hue University, Hue, Vietnam, **5** Vietnam National Cancer Institute, Vietnam National Cancer Hospital, Hanoi, Vietnam, **6** Department of Medical Ethics and Medical Sociology, Hanoi Medical University, Hanoi, Vietnam, **7** Department of Health Economics, Hanoi Medical University, Hanoi, Vietnam

* daidinh.hup@gmail.com (DXD); huongnguyenthanh@hup.edu.vn (HTTN)

**Data Availability Statement:** All relevant data are within the manuscript and its Supporting Information files.

## Abstract

### Background

Patients with human epidermal growth factor receptor 2 (HER2)-positive breast cancer may have poor prognoses and short overall and disease-free survival. Most previous studies focused on assessing the quality of life and health-state utility of the general population of breast cancer patients. The number of studies for HER2-positive breast cancer patients is negligible. This study investigated the health-state utility and its associated factors among Vietnamese patients with HER2-positive breast cancer.

### Methods

We conducted face-to-face interviews with 301 HER2-positive breast cancer patients to collect data. Their health-state utility was measured via the EQ-5D-5L instrument. The Mann-Whitney U and Kruskal-Wallis tests were employed to compare the differences in utility scores between two groups and among three groups or more, respectively. Factors associated with patients' heath-state utility were identified via Tobit regression models.

### Results

Pain/discomfort (56.1%) and anxiety/depression (39.5%) were the two issues that patients suffered from the most, especially among metastatic breast cancer patients. The severity of distress (depression, anxiety, and stress) in patients was relatively mild. Of 301 patients, their average utility score was 0.86±0.17 (range: 0.03–1.00), and the average EQ-visual analogue scale (VAS) score was 69.12±12.60 (range: 30–100). These figures were 0.79 ±0.21 and 65.20±13.20 for 102 metastatic breast cancer patients, significantly lower than those of 199 non-metastatic cancer patients (0.89±0.13 and 71.13±11.78) (*p<0.001*), respectively. Lower health-state utility scores were significantly associated with older age (*p*

**Funding:** The author(s) received no specific funding for this work.

**Competing interests:** The authors declare that they have no competing interests.

= 0.002), lower education level (*p = 0.006*), lower monthly income (*p = 0.036*), metastatic cancer (*p = 0.001*), lower EQ-VAS score (*p<0.001*), and more severe level of distress (*p<0.001*).

## Conclusions

Our findings showed a significant decrement in utility scores among metastatic breast cancer patients. Patients' health-state utility differed by their demographic characteristics (age, education level, and income) and clinical characteristics (stage of cancer and distress). Their utility scores may support further cost-effectiveness analysis in Vietnam.

## Introduction

Globally, it is estimated that there were approximately 2.3 million new cases and 670 thousand deaths by reason of breast cancer in 2022 [1]. This disease was the second most commonly diagnosed cancer (both sexes, all ages: 11.5% of all cancer cases; women: 23.8%) and ranked fourth in the leading causes of cancer deaths [2]. The incidence and mortality rates for this breast disease were 46.8 and 12.7 per 100,000, respectively [2]. In Vietnam, about 24,563 new cases and 10,008 deaths involving breast cancer were reported in 2022. In this country, this breast disease ranked first in the new cancer cases and fourth in the number of cancer deaths (incidence rate: 38.0 per 100,000; mortality rate: 14.7 per 100,000) [2].

The human epidermal growth factor receptor 2 (HER2) receptor, which is a transmembrane glycoprotein with tyrosine kinase activity, plays a role in controlling epithelial cell proliferation and differentiation [3]. The HER2 gene is amplified from two-fold to more than 20-fold among roughly 15–20% of breast cancer patients. The amplification of this gene gives rise to protein expression and the increasing number of receptors at the cell surface, thereby contributing to excessive cellular division and tumor formation [4, 5]. HER2-positive breast cancer is more likely to spread from the breast to other areas of the body (metastasis) and recur than the HER2-negative type of breast cancer. Patients with HER2-positive breast cancer may have poorer prognoses and shorter overall and disease-free survival [6, 7]. Luckily, the discovery of the HER2 gene and the HER2-targeted drugs ushered in a new chapter for patients with both early-stage and metastatic HER2-positive breast cancer. These drugs are recommended for use and feature in numerous countries' standard treatment regimens. Using HER2-targeted drugs (such as trastuzumab, pertuzumab, lapatinib, and neratinib) can make a contribution to improving the treatment effectiveness, reducing the risks of recurrence and mortality, and prolonging survival for HER2-positive breast cancer patients [8–10].

Besides the targets mentioned above, health-related quality of life is also a vital outcome employed to assess the effectiveness and efficacy of cancer treatment therapies. There are a plethora of previous studies conducted to assess the quality of life and health-state utility of breast cancer patients, including several studies in Vietnam [11–13]. However, most of them focused on the general population of breast cancer patients receiving various types of treatment therapies (such as surgery, radiotherapy, and chemotherapy). The number of studies for HER2-positive breast cancer patients (especially patients undergoing systemic treatment) is negligible [14, 15]. In addition, the treatment costs of HER2-targeted drugs are exorbitant. As a result, these medications are of great interest to researchers and policymakers. Measuring the health-related quality of life and health-state utility of HER2-positive breast cancer patients is essential for the health technology assessment of anti-HER2 agents. These data can be used to

compute significant indicators and parameters, thereby providing reliable evidence supporting the decision-making related to health insurance coverage policies. This research was conducted to measure the health-state utility and its associated factors among Vietnamese patients with HER2-positive breast cancer.

## Materials and methods

### Ethical approval

This research was approved by the ethics committee of the Hue University of Medicine and Pharmacy, Hue University, Vietnam (reference number: H2022/495, dated 20th September 2022). The authors also received permission from the board of directors of two hospitals where data were collected. The participants (patients with HER2-positive breast cancer) gave written informed consent before participating in this research. The authors had access to medical records and information that could identify individual participants during the data collection. However, patients' personal information will be kept confidential. The time for patient recruitment and access to medical records was from November 2022 to April 2023.

### Study design and setting

This cross-sectional study was conducted in two hospitals to collect data on patients with HER2-positive breast cancer. The first hospital was the *Vietnam National Cancer Hospital* in northern Vietnam. This is Vietnam's biggest specialized oncology hospital, with approximately 2,000 hospital beds. The second health facility was the *Oncology Center* of *Hue Central Hospital*, a tertiary referral hospital with about 4,500 hospital beds in the central region of Vietnam. This center has about 500 hospital beds and is one of Vietnam's leading cancer treatment facilities.

### Participants and the sample size

The study population was patients with HER2-positive breast cancer, including early (I, II, and III) and late stages (IV). Two standard tests employed to identify whether or not breast cancer is HER2-positive were the ImmunoHistoChemistry (IHC) test and (2) the Fluorescence In Situ Hybridization (FISH) test. The latter is more accurate but expensive and takes longer to have results than the former. Participants' inclusion criteria included (1) female patients aged 18 or older, (2) being diagnosed with HER2-positive breast cancer (having an IHC result of 3 + or an IHC result of 2+ along with a FISH test score of positive), (3) receiving systemic therapy for adjuvant or metastatic treatment (including chemotherapy only, chemotherapy in combination with HER2-targeted therapy, or HER2-targeted therapy with/without hormone therapy), and (4) having at least three cycles of treatment before the time of interviewing. Patients were excluded if they did not concur to participate in this research, did not give written informed consent, and could not finish the questionnaire by virtue of feeling haggard or having psychological issues.

The sample size was computed via the following formula: $n = Z^2_{(1-\alpha/2)}.\sigma^2/(\varepsilon^2.\mu^2)$. With $Z_{(1-\alpha/2)}$ = 1.96 ($\alpha$ = 0.05), $\sigma$ (standard deviation) = 0.22 [13], $\varepsilon$ (margin of error) = 5%, and $\mu$ = 0.74 [13], the minimum sample size (n) was 136 patients. In November 2022, after an initial screening, there were approximately 446 eligible inpatients currently undergoing systemic treatment in two selected hospitals. We strived to approach nearly all of them. However, only 301 patients concurred to participate in this research (response rate: 67.5%). Data collectors (the authors) introduced the study objectives and procedures to patients with HER2-positive breast cancer. Then, they were invited to participate in this research. Those agreeing to take part in

this research gave us their written informed consent and then got a face-to-face interview with data collectors in two selected hospitals. After that, data collectors collected additional data from inpatients' medical records.

## The questionnaire and measurements

Patient's demographic characteristics were collected via a direct interview (including year of birth, residence, education level, job/occupation, marital status, whether or not giving birth, and patient's monthly income). Patient's clinical characteristics and their treatment were gathered from medical records, including the time since the first diagnosis of breast cancer, the stage of cancer (metastatic or non-metastatic), type of metastasis (relapsed or de novo), site of metastasis, comorbidity, menopausal status, hormone receptor (positive or negative), and treatment regimens. In this study, the Depression Anxiety Stress Scale (DASS-10) was utilized to measure patients' level of distress. The validity and reliability of DASS-10 was demonstrated in a previous study [16]. We received the DASS-10 developers' permission to translate and use this scale. The linguistic validation of the DASS-10 consisted of three steps: forward translation, backward translation, and testing. This questionnaire consists of ten questions divided into two subscales (including depression and anxiety-stress). For each question, patients had four answering options, including "never", "sometimes", "often", and "almost always". The total score of a patient can range from 0 to 30 (reflecting the overall distress: (1) mild/subclinical: 6 or less, (2) moderate: 7–12, and (3) severe: 13 or higher) [17].

The primary outcome was patients' health-state utility measured via the EQ-5D-5L questionnaire—one of the multi-attribute utility instruments recommended for use in pharmacoeconomic guidelines in many countries [18]. This instrument comprises two parts: (1) the EQ visual analogue scale (VAS) and (2) the EQ-5D descriptive system. The former is a vertical visual analog scale ranging from 0 (The worst health a patient can imagine) to 100 (The best health a patient can imagine). This scale reflects the patient's health based on their own judgment (self-rated health). The latter includes five dimensions: (1) mobility, (2) self-care, (3) usual activities, (4) pain/discomfort, and (5) anxiety/depression. One dimension consists of five levels of problems: (1) no, (2) slight, (3) moderate, (4) severe, and (5) extreme/unable. Patients were also categorized into two groups for each dimension: no problems if they chose the "no" option and having problems if they chose one of the four remaining options. The Vietnamese version of EQ-5D-5L and the value set for the general population in Vietnam (scoring) are available [19, 20]. The use of this questionnaire was approved by the EuroQoL Research Foundation (request registration number 50956, dated 25 Jul 2022).

## Data analysis

After being collected, data were entered into an Excel file and analyzed with SPSS version 26 and STATA version 15. Mean (standard deviation) or median (25th-75th) was used to report numeric variables (such as patient's age), while categorical variables (such as occupation) were described via numbers and percentages. The relationship between two categorical variables was identified using a Chi-squared or Fisher's exact test. Regarding numeric variables, the Kolmogorov-Smirnov test was used to check the normal distribution of data. By reason of the non-normal distribution, the Mann-Whitney U and Kruskal-Wallis rank-sum tests were employed to compare the differences in utility scores between two groups and among three groups or more, respectively. In addition, factors associated with patients' health-state utility were identified using Tobit regression models that describe the relationship between a truncated or censored continuous variable (a dependent variable) and independent variables, in line with several previous studies [13, 21]. The multivariate model only included independent

variables having p-values<0.05 in univariate analyses. A p-value<0.05 was regarded as statistical significance.

## Results

### The general characteristics of participants

Of 301 participants, there were 199 non-metastatic breast cancer patients (66.1%) and 102 metastatic breast cancer patients (33.9%). On average, patients' age was 51.1±9.5 years old. Roughly 88.0% of patients aged 40 or older. About two-thirds came from rural areas (61.5%) and had an education level of secondary school, high school, or college (71.1%). Most patients got married (95.3%) and gave birth (99.0%). Common occupations were farmers (28.9%), employees/business people (25.2%), and freelancers (24.9%). A quarter had a monthly income of less than two million Vietnam dongs (24.9%), while this figure of nearly two-fifths was higher than three million Vietnam dongs (57.5%). Patients' average income per month was approximately 5.0±4.7 million Vietnam dongs. Between the two patient groups (non-metastatic and metastatic), there were significant differences in education level (*p = 0.021*), occupation (*p<0.001*), giving birth (*p = 0.038*), and patient's income per month (*p = 0.009*) (Table 1).

   Of 102 metastatic breast cancer patients, nearly a third had de novo metastatic cancer. More than three-quarters had visceral and/or central nervous system metastasis. On average, the time since the first diagnosis of breast cancer among all 301 patients was 20.6±32.3 months (non-metastatic patients: 7.7±4.5 months and metastatic patients: 45.8±45.6 months). Among all patients, 31.2% had at least one comorbidity, and 57.8% were menopausal women. The percentage of patients with hormone receptor-positive breast cancer (49.2%) was relatively equal to that of patients with hormone receptor-negative breast cancer (50.8%). Regarding the treatment regimens, nearly half of non-metastatic breast cancer patients (47.2%) were given HER2-targeted therapy only or in combination with hormone therapy. Meanwhile, a majority of metastatic breast cancer patients received chemotherapy only (53.9%) or in combination with HER2-target therapy (32.4%). Between the two patient groups (non-metastatic and metastatic), there were significant differences in the time since the first diagnosis (*p<0.001*), menopausal status (*p = 0.048*), hormone receptor (*p = 0.001*), and received treatment regimens (*p<0.001*) (Table 2).

### Patients' health profile across five health dimensions of EQ 5D-5L and DASS-10

Among all 301 patients, pain/discomfort (56.1%) and anxiety/depression (39.5%) were the two issues that many patients suffered from most. In comparison with the non-metastatic group, the percentages of patients with metastatic breast cancer having problems involving usual activities, pain/discomfort, and anxiety/depression were significantly higher (*p = 0.012*, *p<0.001*, and *p<0.001*, respectively). Regarding these three dimensions, the number of metastatic breast cancer patients having severe or extreme problems was also higher than that of the non-metastatic patient group (usual activities: 5.9% and 0.0%; pain/discomfort: 9.8% and 4.0%; and anxiety/depression: 7.9% and 4.0%, respectively). Among all 301 patients with HER2-positive breast cancer, their average utility score was 0.86±0.17 (range: 0.03–1.00), and the average EQ-VAS score was 69.12±12.60 (range: 30–100). These scores of 102 metastatic cancer patients (0.79±0.21 and 65.20±13.20) were significantly lower than those of 199 non-metastatic cancer patients (0.89±0.13 and 71.13±11.78) (*p<0.001* and *p<0.001*), respectively. In addition, the average DASS-10 score of the former was slightly higher than that of the latter, but this difference was insignificant (*p = 0.245*) (Table 3, Fig 1).

**Table 1. Demographic characteristics of participants (n = 301 patients).**

| Demographic characteristics | | Patients n (%) | | | p-value |
|---|---|---|---|---|---|
| | | **All** | **Non-metastatic** | **Metastatic** | |
| **Age** | <40 | 36 (12.0) | 24 (12.1) | 12 (11.8) | 0.070 |
| | 40–49 | 88 (29.2) | 66 (33.2) | 22 (21.6) | |
| | 50–59 | 116 (38.5) | 76 (38.2) | 40 (39.2) | |
| | ≥60 | 61 (20.3) | 33 (16.6) | 28 (27.5) | |
| **Residence** | Urban | 116 (38.5) | 75 (37.7) | 41 (40.2) | 0.672 |
| | Rural | 185 (61.5) | 124 (62.3) | 61 (59.8) | |
| **Education level** | Primary school or lower | 15 (5.0) | 8 (4.1) | 7 (6.9) | **0.021** |
| | Secondary school | 102 (33.9) | 59 (29.6) | 43 (42.2) | |
| | High school or college | 139 (37.2) | 95 (47.7) | 44 (43.1) | |
| | University or higher | 45 (15.0) | 37 (18.6) | 8 (7.8) | |
| **Job (occupation)** | Employment or business | 76 (25.2) | 64 (32.2) | 12 (11.8) | **<0.001** |
| | Agriculture | 87 (28.9) | 44 (22.1) | 43 (42.2) | |
| | Housework | 22 (7.3) | 13 (6.5) | 9 (8.8) | |
| | Freelancers | 75 (24.9) | 51 (25.6) | 24 (23.5) | |
| | Retired | 41 (13.6) | 27 (13.6) | 14 (13.7) | |
| **Marital status** | Unmarried or divorced | 14 (4.7) | 7 (3.5) | 7 (6.9) | 0.247 |
| | Married | 287 (95.3) | 192 (96.5) | 95 (93.1) | |
| **Gave birth** | No | 3 (1.0) | 0 (0.0) | 3 (2.9) | **0.038** |
| | Yes | 298 (99.0) | 199 (100.0) | 99 (97.1) | |
| **Patient monthly income (mVND)*** | No income | 49 (16.3) | 28 (14.1) | 21 (20.6) | **0.009** |
| | ≤2 | 26 (8.6) | 15 (7.5) | 11 (10.8) | |
| | >2 to 3 | 48 (15.9) | 25 (12.6) | 23 (22.5) | |
| | >3 | 173 (57.5) | 128 (64.3) | 45 (44.1) | |
| | No answering | 5 (1.7) | 3 (1.5) | 2 (2.0) | |

*1 mVND (million Vietnam dongs) = 41.27 US dollars.

**Factors associated with the health-state utility of patients with HER2-positive breast cancer.** The average utility score of patients aged 60 or older was 0.80±0.19, significantly lower than that of other age groups (*p<0.001*). This figure for illiterate patients and those only graduating from a primary school (0.69±0.25) was also significantly lower than that of patients with higher education levels (*p<0.001*). With regard to patients' occupations, the average utility score of employees and business people was the highest (0.93±0.10), while that of house-workers was the lowest (0.76±0.18) (*p<0.001*). Among income groups, patients with more than three million Vietnam dongs per month had the highest average utility score (0.90±0.11, *p<0.001*). Regarding clinical characteristics, lower average utility scores were witnessed among patients with a longer time since the first diagnosis of breast cancer (*p = 0.002*), meno-pausal women (*p = 0.001*), and patients receiving chemotherapy in combination with HER2--targeted therapy (*p = 0.007*) (Table 4).

As per the results of the Tobit regression model, the health-state utility of patients with HER-2 positive breast cancer was significantly associated with their age, education level, monthly income, stage of cancer (non-metastatic or metastatic), EQ-VAS score, and DASS-10 score. The higher the patient's age and the DASS-10 score, the lower the utility scores (negative correlations). For each one-unit increase in patient's age (one year) and DASS-10 score (one score), their utility scores decreased by 0.005 (*p = 0.002*) and 0.007 (*p<0.001*), respectively. In contrast, there was a positive correlation between the EQ-VAS scores and health-state utility

**Table 2. Clinical characteristics of participants (n = 301 patients).**

| Clinical characteristics | | Patients n (%) | | | p-value |
|---|---|---|---|---|---|
| | | All | Non-metastatic | Metastatic | |
| **Stage of cancer (tumor stage)** | I, II, or III | 199 (69.1) | 199 (100.0) | - | - |
| | IV | 102 (33.9) | - | 102 (100.0) | |
| **Type of metastasis** | Relapsed | - | - | 69 (67.6) | - |
| | De novo | - | - | 33 (32.4) | |
| **Site of metastasis** | Visceral or central nervous system | - | - | 78 (76.5) | - |
| | Non-visceral | - | - | 24 (23.5) | |
| **The time since the first diagnosis of breast cancer (months)** | ≤12 | 191 (63.5) | 165 (82.9) | 26 (25.5) | <0.001 |
| | 13–36 | 66 (21.9) | 34 (17.1) | 32 (31.4) | |
| | 37–60 | 21 (7.0) | 0 (0.0) | 21 (20.6) | |
| | >60 | 23 (7.6) | 0 (0.0) | 23 (22.5) | |
| **Comorbidity** | Yes | 74 (24.6) | 53 (26.6) | 21 (20.6) | 0.249 |
| | No | 227 (75.4) | 146 (73.4) | 81 (79.4) | |
| **Menopause** | Yes | 174 (57.8) | 107 (53.8) | 67 (65.7) | **0.048** |
| | No | 127 (42.2) | 92 (46.2) | 35 (34.3) | |
| **Hormone receptor** | Positive | 148 (49.2) | 111 (55.8) | 37 (36.3) | **0.001** |
| | Negative | 153 (50.8) | 88 (44.2) | 65 (63.7) | |
| **Treatment regimen** | Chemotherapy only | 115 (38.2) | 60 (30.2) | 55 (53.9) | <0.001 |
| | Chemotherapy + HER2-targeted therapy | 78 (25.9) | 45 (22.6) | 33 (32.4) | |
| | HER2-targeted therapy with or without hormone therapy | 108 (35.9) | 94 (47.2) | 14 (13.7) | |

HER2: human epidermal growth factor receptor 2.

(beta = 0.007, *p<0.001*). Patients' utility scores also increased by 0.008 for each one-unit monthly income increase (one million Vietnam dongs) (*p = 0.036*). In addition, higher utility scores were found among patients with an education level of university or higher (*p = 0.006*) and non-metastatic breast cancer patients (*p = 0.001*) (Table 5).

## Discussion

This study was conducted to measure HER2-positive breast cancer patients' health-state utility and associated factors in Vietnam using the EQ-5D-5L instrument. Our findings showed that the average utility score of 301 patients with HER2-positive breast cancer was 0.86±0.17. This figure among non-metastatic breast cancer patients (0.89±0.13) was significantly higher than that of the metastatic cancer patient group (0.79±0.21). The utility scores were positively correlated with patients' monthly income and EQ-VAS score but negatively correlated with their age and level of distress (DASS-10 score). In addition, education level was another factor associated with the health-state utility of these patients.

**Table 3. Patients' health profile across five health dimensions of EQ 5D-5L and DASS-10.**

| Dimensions | | Patients n (%) | | | p-value |
|---|---|---|---|---|---|
| | | All | Non-metastatic | Metastatic | |
| **Mobility** | No problems | 233 (77.4) | 160 (80.4) | 73 (71.6) | 0.083 |
| | Having problems | 68 (22.6) | 39 (19.6) | 29 (28.4) | |
| **Self-care** | No problems | 265 (88.0) | 180 (90.5) | 85 (83.3) | 0.072 |
| | Having problems | 36 (12.0) | 19 (9.5) | 17 (16.7) | |
| **Usual activities** | No problems | 260 (86.4) | 179 (89.9) | 81 (79.4) | **0.012** |
| | Having problems | 41 (13.6) | 20 (10.1) | 21 (20.6) | |
| **Pain/discomfort** | No problems | 132 (43.9) | 105 (52.8) | 27 (26.5) | **<0.001** |
| | Having problems | 169 (56.1) | 94 (47.2) | 75 (73.5) | |
| **Anxiety/depression** | No problems | 182 (60.5) | 136 (68.3) | 46 (45.1) | **<0.001** |
| | Having problems | 119 (39.5) | 63 (31.7) | 56 (54.9) | |
| **Utility score** (mean±SD) | | 0.86±0.17 | 0.89±0.13 | 0.79±0.21 | **<0.001** |
| **EQ-VAS score** (mean±SD) | | 69.12±12.60 | 71.13±11.78 | 65.20±13.20 | **<0.001** |
| **DASS-10 score** (mean±SD) | | 5.04±6.38 | 4.82±6.34 | 5.47±6.47 | 0.245 |

EQ-VAS: EQ visual analogue scale, DASS-10: Depression Anxiety Stress Scale, SD: standard deviation.

## Patients' health profile across five health dimensions of EQ 5D-5L and DASS-10

Pain/discomfort and anxiety/depression were two common health issues that many Vietnamese patients with HER2-positive breast cancer suffered from (56.1% and 39.5%, respectively). The number of patients with severe or extreme problems with these two dimensions was also the highest. By contrast, not many patients had problems involving self-care (12.0%). These results were in line with the findings of numerous previous studies in Brazil, China, Ethiopia, Korea, and Singapore [13, 14, 22–26]. However, anxiety/depression was the most common health issue among breast cancer patients in India and Indonesia, followed by pain/discomfort [27, 28]. In the United Kingdom, usual activities and pain/discomfort were two common issues of HER2-positive breast cancer patients [15]. The differences in the findings of studies can be explained by the differences in studying time, location, stage of cancer, and the sample. Besides, in spite of the high number of Vietnamese breast cancer patients having problems with anxiety/depression, the severity of this health dimension was relatively mild (according to the DASS-10 scores). Nonetheless, both physical and mental health should be paid attention to when taking care of breast cancer patients.

For all five dimensions, the percentage of patients having problems in the metastatic group was higher than that in the non-metastatic group. However, the differences were only significant in three dimensions, including usual activities, pain/discomfort, and anxiety/depression. It is perfectly understandable that the severity of these health dimensions among the former was greater than that of the latter. Two studies in Brazil and China reported similar results when patients with metastatic breast cancer suffered from more health problems than the non-metastatic group [14, 24]. However, in these two studies, the differences were statistically significant in all five dimensions. In the United Kingdom, the differences among patient groups were statistically significant in mobility, self-care, and usual activities. The other two dimensions (including pain/discomfort and anxiety/depression) did not find any significant differences [15]. The findings from a study in the United States also showed that the spiritual and emotional quality of life outcomes among advanced breast cancer women were not significantly different as per metastatic status [29]. The differences in location, patient selection

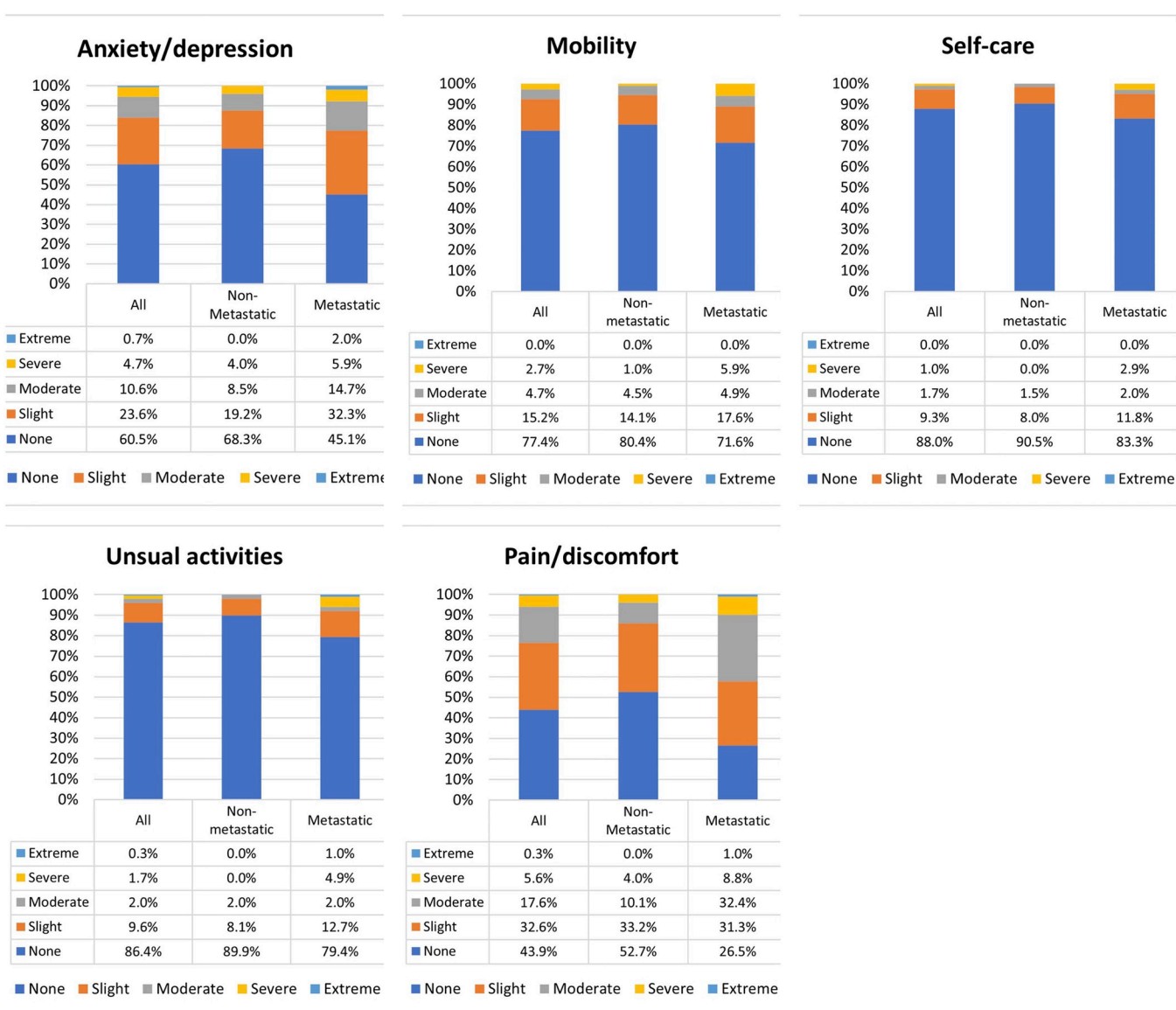

**Fig 1. The proportion of patient responses by the extent of problems reported in each EQ-5D-5L domain among non-metastatic and metastatic HER2-positive breast cancer patients.**

criteria, year of survey, patient's income, and healthcare systems may be possible rationales behind the differences in the findings among countries.

## Patients' EQ-VAS scores

The average EQ-VAS score of Vietnamese patients with HER2-positive breast cancer was 69.12, in line with the result of a study in Ethiopia (69.9) [26]. Our result was slightly higher than the result of a previous study in Vietnam (64.9) [13] but far lower than the result of a study among Chinese patients (80.0) [23]. This difference may spring from the differences between the sample, location, studying time, and treatment therapies. Regarding the stages of cancer, in our research, there was a significant difference in the EQ-VAS scores between patients with metastatic breast cancer (65.20) and those with non-metastatic breast cancer

**Table 4. Breast cancer patients' average utility scores by their demographic and clinical characteristics.**

| Independent variables | | Mean±SD | Min-Max | Median (25th-75th) | p-value |
|---|---|---|---|---|---|
| **Demographic characteristics** | | | | | |
| **Age** | <40 | 0.88±0.20 | 0.03–1.00 | 0.95 (0.85–1.00) | **<0.001** |
| | 40–49 | 0.89±0.12 | 0.56–1.00 | 0.92 (0.84–1.00) | |
| | 50–59 | 0.85±0.17 | 0.03–1.00 | 0.88 (0.80–1.00) | |
| | ≥60 | 0.80±0.19 | 0.03–1.00 | 0.85 (0.74–0.92) | |
| **Residence** | Urban | 0.86±0.18 | 0.03–1.00 | 0.92 (0.78–1.00) | 0.302 |
| | Rural | 0.85±0.16 | 0.03–1.00 | 0.89 (0.80–1.00) | |
| **Education level** | Primary school or lower | 0.69±0.25 | 0.03–0.94 | 0.74 (0.58–0.87) | **<0.001** |
| | Secondary school | 0.83±0.18 | 0.03–1.00 | 0.85 (0.78–0.93) | |
| | High school or college | 0.87±0.15 | 0.03–1.00 | 0.92 (0.80–1.00) | |
| | University or higher | 0.94±0.10 | 0.46–1.00 | 1.00 (0.90–1.00) | |
| **Job (occupation)** | Employment or business | 0.93±0.10 | 0.46–1.00 | 0.98 (0.89–1.00) | **<0.001** |
| | Agriculture | 0.84±0.17 | 0.03–1.00 | 0.85 (0.78–0.94) | |
| | Housework | 0.76±0.18 | 0.37–1.00 | 0.85 (0.65–0.88) | |
| | Freelancers | 0.81±0.22 | 0.03–1.00 | 0.85 (0.74–1.00) | |
| | Retired | 0.89±0.10 | 0.68–1.00 | 0.92 (0.84–1.00) | |
| **Marital status** | Unmarried or divorced | 0.81±0.16 | 0.37–1.00 | 0.85 (0.74–0.94) | 0.099 |
| | Married | 0.86±0.17 | 0.03–1.00 | 0.92 (0.80–1.00) | |
| **Giving birth** | No | 0.91±0.15 | 0.73–1.00 | 1.00 (0.73–1.00) | 0.498 |
| | Yes | 0.86±0.17 | 0.03–1.00 | 0.89 (0.80–1.00) | |
| **Patient monthly income (mVND)*** | No income | 0.81±0.18 | 0.07–1.00 | 0.85 (0.73–0.92) | **<0.001** |
| | ≤ 2 | 0.80±0.18 | 0.18–1.00 | 0.81 (0.74–0.92) | |
| | > 2 to 3 | 0.79±0.25 | 0.03–1.00 | 0.85 (0.75–0.94) | |
| | > 3 | 0.90±0.11 | 0.36–1.00 | 0.92 (0.85–1.00) | |
| **Clinical characteristics** | | | | | |
| **The time since the first diagnosis of breast cancer (months)** | ≤12 | 0.88±0.17 | 0.03–1.00 | 0.92 (0.85–1.00) | **0.002** |
| | 13–36 | 0.83±0.17 | 0.18–1.00 | 0.87 (0.78–0.94) | |
| | 37–60 | 0.81±0.13 | 0.57–1.00 | 0.85 (0.70–0.90) | |
| | >60 | 0.82±0.20 | 0.03–1.00 | 0.85 (0.74–1.00) | |
| **Comorbidity** | Yes | 0.86±0.19 | 0.03–1.00 | 0.92 (0.85–1.00) | 0.393 |
| | No | 0.86±0.16 | 0.03–1.00 | 0.89 (0.78–1.00) | |
| **Menopause** | Yes | 0.84±0.17 | 0.03–1.00 | 0.85 (0.78–0.94) | **0.001** |
| | No | 0.88±0.17 | 0.03–1.00 | 0.92 (0.85–1.00) | |
| **Hormone receptor** | Positive | 0.87±0.17 | 0.03–1.00 | 0.92 (0.83–1.00) | 0.093 |
| | Negative | 0.85±0.17 | 0.03–1.00 | 0.85 (0.78–1.00) | |
| **Treatment regimen** | Chemotherapy only | 0.87±0.13 | 0.03–1.00 | 0.85 (0.81–0.96) | **0.007** |
| | Chemotherapy + targeted therapy | 0.80±0.23 | 0.03–1.00 | 0.85 (0.72–1.00) | |
| | Targeted therapy with or without hormone therapy | 0.89±0.14 | 0.37–1.00 | 0.92 (0.85–1.00) | |

*1 mVND (million Vietnam dongs) = 41.27 US dollars. SD: standard deviation.

(71.13). Our findings were similar to the results of a study in the United Kingdom (patients receiving treatment for metastatic breast cancer: 65.82 and patients currently undergoing treatment for early breast cancer: 72.74) [15]. In comparison with non-metastatic breast cancer patients, the higher EQ-VAS scores of patients with metastatic breast cancer were also reported in Brazil, China, India, and Indonesia [14, 23, 24, 27, 28].

**Table 5. Factors associated with the health-state utility of patients with HER2-positive breast cancer (Tobit regression analyses).**

| Independent variable | | Univariate | | Multivariate | |
|---|---|---|---|---|---|
| | | coef. | p-value | a.coef. | p-value |
| Age (a numeric variable) | | -0.006 | **<0.001** | -0.005 | **0.002** |
| Education level (ref: primary school or lower) | Secondary school | 0.166 | **0.005** | 0.055 | 0.217 |
| | High school or college | 0.230 | **<0.001** | 0.086 | 0.060 |
| | University or higher | 0.364 | **<0.001** | 0.161 | **0.006** |
| Job (occupation) (ref: Housework) | Employment or business | 0.247 | **<0.001** | -0.053 | 0.299 |
| | Agriculture | 0.100 | **0.059** | 0.059 | 0.143 |
| | Freelancers | 0.076 | 0.144 | -0.045 | 0.280 |
| | Retired | 0.154 | **0.007** | 0.025 | 0.602 |
| Patient monthly income (mVND)* (a numeric variable) | | 0.019 | **<0.001** | 0.008 | **0.036** |
| Menopause (ref: No) | Yes | -0.075 | **0.007** | 0.029 | 0.288 |
| Metastasis (ref: No) | Yes | -0.149 | **<0.001** | -0.077 | **0.001** |
| Treatment regimen (ref: HER2-targeted therapy with or without hormone therapy) | Chemotherapy only | -0.050 | 0.104 | 0.018 | 0.475 |
| | Chemotherapy + HER2-targeted therapy | -0.120 | **<0.001** | -0.047 | 0.074 |
| EQ-VAS score (a numeric variable) | | 0.009 | **<0.001** | 0.007 | **<0.001** |
| **DASS-10 score** (a numeric variable) | | -0.012 | **<0.001** | -0.007 | **<0.001** |

*1 mVND (million Vietnam dongs) = 41.27 US dollars. Ref: reference, coef.: coefficient, a.coef.: adjusted coefficient, EQ-VAS: EQ visual analogue scale, DASS-10: Depression Anxiety Stress Scale.

The multicollinearity was checked using the Variance Inflation Factor (VIF) for each independent variable. In the multivariate model, all VIF values were lower than 2.5.

## Health-state utility and associated factors

Of all 301 Vietnamese patients with HER2-positive breast cancer, their average utility score was 0.86, consistent with the results of studies in China, Ethiopia, and India [23, 26, 27, 30] but higher than the findings of a study in Singapore [25]. In comparison with the utility score of breast cancer patients from a previous study in Vietnam in 2019, our result was higher (0.74 and 0.86, respectively) [13]. Used treatment therapies can be the rationale behind the difference between the two studies. In this study, we only recruited patients receiving systemic treatment. Meanwhile, in the study in 2019, breast cancer patients could receive various kinds of treatment therapies, such as lumpectomy, mastectomy, and breast reconstruction surgery.

The stage of cancer/metastatic status is a crucial factor affecting treatment targets and regimens for breast cancer patients. The average utility scores of metastatic and non-metastatic breast cancer patients in Vietnam were 0.79 and 0.89, higher than those of HER2-positive breast cancer patients in the United Kingdom (0.70 and 0.81), respectively [15]. For both metastatic and non-metastatic cancer groups, there were also differences in the utility scores of patients among countries (such as the Netherlands and Sweden [31], Korea [32], China [23, 33], Indonesia [28]), but the gaps were not huge. In addition, previous studies showed that the health-related quality of life and health-state utility of patients with metastatic breast cancer were significantly lower than those of patients with early-stage and non-metastatic breast cancer [34, 35]. As a result, it is necessary to have screening programs to detect and diagnose breast cancer at an early stage and have suitable healthcare programs prioritized for late-stage cancer patients.

In this study, three demographic characteristics of Vietnamese patients associated with their health-state utility were age, education level, and monthly income. In Asia, breast cancer patients receiving chemotherapy, having comorbidities, less social support, and more unmet needs had poorer health-related quality of life [34]. In low- and middle-income countries in

Asia, breast cancer patients' health-related quality of life was associated with their age, marital status, education level, income, stage of the tumor, method, treatment duration, and lifestyle [36]. Similar to our results, the poor health-related quality of life and health-state utility of breast cancer patients with old age, low education level, and low income per month were also found among patients in other countries, such as China, Korea, and India [13, 24, 27, 32, 37, 38]. However, in several studies in China and Japan, younger patients had lower health-related quality of life by reason of being under a lot of pressure and stress [39–41]. With regard to patients' monthly income and education level, our findings were in line with the results of a systematic review: those with higher income and education levels had higher health-related quality of life [36]. Patients with low income and low education levels may be unable to afford the treatment costs and not comprehend the crucial role of treatment adherence, thereby lowering the treatment's effectiveness and efficacy. For patients living on the breadline, health insurance can play an essential role in their treatment process. Sadly, in Vietnam, health insurance does not cover the treatment costs of nearly all HER2-targeted drugs, except for trastuzumab (with a reimbursement rate of 60%). To improve HER2-positive breast cancer patients' health outcomes and health-related quality of life, the government, policymakers, and other relevant parties (especially in low- and middle-income countries) should have practical policies and solutions to support patients in their long-term treatment period.

Besides the four factors above, the health-state utility of HER2-positive breast cancer patients in Vietnam was also associated with their EQ-VAS score and level of distress (DASS-10 score). Among patients with metastatic breast cancer in Germany, their EQ-VAS scores were significantly associated with single EQ-5D-5L items and the total score [42]. Furthermore, breast cancer patients usually suffer from mental health and psychological issues [43–45]. Previous studies demonstrated that breast cancer patients suffering from depression, anxiety, and stress also had lower health-related quality of life [11, 46, 47]. These burdens highlighted the essential roles of physical and mental health care for breast cancer patients, especially for the metastatic cancer group. Improving patients' physical and mental health is of paramount importance [48].

## Strengths and limitations

This is the first study conducted in Vietnam to measure the health-state utility of 301 HER2-positive breast cancer patients in two health facilities. The reliability and validity of the instruments were demonstrated. Information about patients' health profiles, the division of patients into metastatic and non-metastatic cancer groups to analyze data, and factors associated with their health-related quality of life can contribute to developing healthcare programs and supporting their treatment process. In addition, the utility scores computed from the EQ-5D-5L instrument for Vietnamese patients receiving systemic treatment can be used in the health technology assessment of HER2-target drugs, especially in case of the heterogeneity in the utility scores of breast cancer patients among countries around the world. Besides the strengths above, our research has several limitations. First, since patients with HER2-positive breast cancer only account for approximately 15–20% of breast cancer patients, our sample size was not high. Second, collecting data via patient interviews can bring several biases, such as recall bias. Furthermore, by virtue of being a cross-sectional study, our results only reflect patients' information at a point in time, and the causal relationship between the health-state utility and independent factors cannot be determined.

## Conclusions

Many patients with HER2-positive breast cancer had problems involving pain/discomfort and anxiety/depression, especially among metastatic breast cancer patients. Patients' utility scores

were relatively high, and their health issues involving depression, anxiety, and stress were relatively mild. Factors associated with patients' health-state utility included their age, education level, monthly income, metastatic status, EQ-VAS score, and level of distress. Both physical and mental health should be paid attention to when taking care of breast cancer patients. Our findings provide important information that can be used in the health technology assessment of HER2-directed therapy and make a contribution to supporting patients during their long treatment period.

## Supporting information

**S1 File. Data file (raw data).**
(XLSX)

## Acknowledgments

We would like to thank the Board of Directors of the Vietnam National Cancer Hospital and the Hue Central Hospital, doctors, nurses, and patients for their participation and support during the data collection process.

## Author Contributions

**Conceptualization:** Tram Nguyen Nguyet Luu, Dai Xuan Dinh, Thinh Xuan Tran, Thang Binh Tran, Huong Thanh Tran, Kiet Huy Tuan Pham, Huong Thi Thanh Nguyen.

**Data curation:** Tram Nguyen Nguyet Luu, Huong Thi Thanh Nguyen.

**Formal analysis:** Tram Nguyen Nguyet Luu, Dai Xuan Dinh, Thinh Xuan Tran, Thang Binh Tran.

**Investigation:** Tram Nguyen Nguyet Luu, Thinh Xuan Tran, Huong Thi Thanh Nguyen.

**Methodology:** Tram Nguyen Nguyet Luu, Dai Xuan Dinh, Thang Binh Tran, Huong Thanh Tran, Kiet Huy Tuan Pham, Huong Thi Thanh Nguyen.

**Project administration:** Dai Xuan Dinh, Huong Thanh Tran, Kiet Huy Tuan Pham, Huong Thi Thanh Nguyen.

**Supervision:** Dai Xuan Dinh, Huong Thanh Tran, Kiet Huy Tuan Pham, Huong Thi Thanh Nguyen.

**Validation:** Dai Xuan Dinh.

**Visualization:** Tram Nguyen Nguyet Luu, Dai Xuan Dinh.

**Writing – original draft:** Tram Nguyen Nguyet Luu, Dai Xuan Dinh.

**Writing – review & editing:** Tram Nguyen Nguyet Luu, Dai Xuan Dinh, Thinh Xuan Tran, Thang Binh Tran, Huong Thanh Tran, Kiet Huy Tuan Pham, Huong Thi Thanh Nguyen.

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
