## [Decision Letter · Decision Letter 0]

13 Mar 2024

PONE-D-23-42055Health-state utility of patients with HER2-positive breast cancer in Vietnam: a multicenter cross-sectional studyPLOS ONE

Dear Dr. Dinh,

Thank you for submitting your manuscript to PLOS ONE. After careful consideration, we feel that it has merit but does not fully meet PLOS ONE’s publication criteria as it currently stands. Therefore, we invite you to submit a revised version of the manuscript that addresses the points raised during the review process.

We look forward to receiving your revised manuscript.

Kind regards,

Ravishankar Jayadevappa

Academic Editor

PLOS ONE

Journal Requirements:

2. In the online submission form, you indicated that The dataset underpinning findings in this manuscript contain potentially identifying and sensitive patient information. However, the datasets used and/or analyzed during the current study are available from the corresponding author upon reasonable request.

Additional Editor Comments:

Authors are embarked on an important study in the area of utility and cost-effectiveness of breast cancer treatment. The articles require careful revision in terms of editing and proof reading. Also, some statements are unsupported, appropriate references should be used.

Reviewers' comments:

Reviewer's Responses to Questions

**Comments to the Author**

1. Is the manuscript technically sound, and do the data support the conclusions?

Reviewer #1: Yes

Reviewer #2: Yes

2. Has the statistical analysis been performed appropriately and rigorously? 

Reviewer #1: Yes

Reviewer #2: Yes

3. Have the authors made all data underlying the findings in their manuscript fully available?

Reviewer #1: Yes

Reviewer #2: No

4. Is the manuscript presented in an intelligible fashion and written in standard English?

Reviewer #1: No

Reviewer #2: Yes

5. Review Comments to the Author

Reviewer #1: The findings of this study based on breast cancer patients in Viet Nam can add value as the literature suggests that there are differences in health-related quality of life of the patients across countries. However, some revisions and clarifications would benefit the paper. Some of suggestions are listed as follows.

Abstract-Objective

Please add a little background for conducting this study.

Introduction

Lines 72-73, “ The number of studies … is negligible.” This statement seems to indicate there are existing studies. If that is the case, please cite some references and explain why there is a need for this current study based on the shortcomings of the previous studies.

Methods

Lines 95-96, do standard hospital beds differ from inpatient beds? Please explain the difference as succinctly as possible. Also the authors may want to leave out the detail.

Lines 109-111, please rewrite the sentence, which has grammatical and punctuation errors.

Acronyms come after, not before, their full terms. However, the authors switched the order. For example, it should be the Depression Anxiety Stress Scale (DASS-10), not the other way around.

In lines 121-122, do the authors mean to include both personal and household income? This seems unnecessary and problematic. However, the results section seems to indicate that only the personal income was entered into the regression model. Please clarify.

Line 135, in “… use in national health technology guidelines,” please be more specific which nation recommends using EQ-5D-5L. This statement can be misinterpreted as Vietnamese government’s guidelines.

The top line of Table 2 is mislabeled. It should not be “demographic characteristics.”

In Table 3 and in line 250, there should be mobility instead of morbidity.

In general, this manuscript would benefit from thorough spell and grammatical checks.

Discussion

Lines 245-253, please try to explain potential reasons for similarities between Viet Nam and Brazil/China and for differences between Viet Nam and the U.K/U.S.

Lines 270-272 seems to hint a rationale for conducting this present study which relied on patients on pharmacological treatment only whereas the 2019 study derived utility scores from Vietnamese patients who underwent surgery as well. This point could have been used to reinforce the need for the current study in the introduction section.

Reviewer #2: This is a sound piece of scientific research highlighting population characteristics of HER2 positive breast cancer patients in Vietnam. The authors do not make their data available due to potentially identifiable information, which is reasonable in the context of this manuscript.

6. PLOS authors have the option to publish the peer review history of their article (what does this mean?). If published, this will include your full peer review and any attached files.

Reviewer #1: No

Reviewer #2: No

---

## [Author Response · Author response to Decision Letter 0]

22 Mar 2024

First and foremost, we want to express our deep gratitude to the editors and reviewers who gave us valuable and practical advice. We carefully read your comments and revised our manuscript. Our responses are written in blue color. 

Thank you so much for your assistance.

..........................................................................................................................

REVIEWER REPORTS

Reviewer #1: The findings of this study based on breast cancer patients in Viet Nam can add value as the literature suggests that there are differences in health-related quality of life of the patients across countries. However, some revisions and clarifications would benefit the paper. Some of suggestions are listed as follows.

Abstract-Objective

Please add a little background for conducting this study.

We added some information about reasons for conducting our study in the Abstract.

Introduction

Lines 72-73, “ The number of studies … is negligible.” This statement seems to indicate there are existing studies. If that is the case, please cite some references and explain why there is a need for this current study based on the shortcomings of the previous studies.

We cited two previous studies about HER2-positive breast cancer patients in the UK and Brazil.

Methods

Lines 95-96, do standard hospital beds differ from inpatient beds? Please explain the difference as succinctly as possible. Also the authors may want to leave out the detail.

We revised this issue. They are similar. We used a general term (hospital bed) for these data.

Lines 109-111, please rewrite the sentence, which has grammatical and punctuation errors.

We revised this sentence.

Acronyms come after, not before, their full terms. However, the authors switched the order. For example, it should be the Depression Anxiety Stress Scale (DASS-10), not the other way around.

We revised this issue.

In lines 121-122, do the authors mean to include both personal and household income? This seems unnecessary and problematic. However, the results section seems to indicate that only the personal income was entered into the regression model. Please clarify.

We removed the variable “household monthly income” in the whole manuscript. We collected data on both patient and household monthly income. Sadly, there were many missing values about household monthly income (51 out of 301 participants). Numerous missing values can lower the quality of regression models. As a consequence, we removed this variable.

Line 135, in “… use in national health technology guidelines,” please be more specific which nation recommends using EQ-5D-5L. This statement can be misinterpreted as Vietnamese government’s guidelines.

We revised this issue. 

In the article cited, “Thirty-four pharmacoeconomic guidelines were included for review. Multi-attribute utility instruments named for use in cost-utility analysis: EQ-5D (n = 29 guidelines), the SF-6D (n = 11), HUI (n = 10), QWB (n = 3), AQoL (n = 2), CHU9D (n = 1). EQ-5D was a preferred multi-attribute utility instrument in 15 guidelines.”

Kennedy-Martin M, Slaap B, Herdman M, van Reenen M, Kennedy-Martin T, Greiner W, et al. Which multi-attribute utility instruments are recommended for use in cost-utility analysis? A review of national health technology assessment (HTA) guidelines. Eur J Health Econ. 2020;21(8):1245-1257. doi: 10.1007/s10198-020-01195-8.

The top line of Table 2 is mislabeled. It should not be “demographic characteristics.”

In Table 3 and in line 250, there should be mobility instead of morbidity.

In general, this manuscript would benefit from thorough spell and grammatical checks.

We corrected these mistakes. 

We endeavoured to check linguistic issues. Thank you for this helpful comment.

Discussion

Lines 245-253, please try to explain potential reasons for similarities between Viet Nam and Brazil/China and for differences between Viet Nam and the U.K/U.S.

We added some reasons for these points.

Lines 270-272 seems to hint a rationale for conducting this present study which relied on patients on pharmacological treatment only whereas the 2019 study derived utility scores from Vietnamese patients who underwent surgery as well. This point could have been used to reinforce the need for the current study in the introduction section.

Thank you for this helpful comment.

If there is anything inappropriate or inadequate, please let us know. Thank you so much for your valuable comments.

Best regards.

Reviewer #2: This is a sound piece of scientific research highlighting population characteristics of HER2 positive breast cancer patients in Vietnam. The authors do not make their data available due to potentially identifiable information, which is reasonable in the context of this manuscript.

The data file is included in the Supporting Information files. We removed the names of patients and shared de-identified or anonymized data to make a small contribution to the scientific integrity.

Abstract: Conclusions are too broad, please revise this to directly reflect the study results

We revised the Conclusion in the Abstract.

Lines 47 – 49: Insert reference

We revised this issue.

Lines 47 – 53: newer data?

Data in 2022 were added.

Line 111 – 114: time during which of data collection occurred for individual patient should be reported, (pre-treatment or post-treatment).

We added some information about the time during which of data collection occurred.

If there is anything inappropriate or inadequate, please let us know. Thank you so much for your valuable comments.

Best regards.

---

## [Decision Letter · Decision Letter 1]

10 Apr 2024

PONE-D-23-42055R1Health-state utility of patients with HER2-positive breast cancer in Vietnam: a multicenter cross-sectional studyPLOS ONE

Dear Dr. Dinh, Thank you for submitting your manuscript to PLOS ONE. After careful consideration, we feel that it has merit but does not fully meet PLOS ONE’s publication criteria as it currently stands. Therefore, we invite you to submit a revised version of the manuscript that addresses the points raised during the review process. Also, please use professional editing services.

We look forward to receiving your revised manuscript.

Kind regards,

Ravishankar Jayadevappa

Academic Editor

PLOS ONE

Journal Requirements:

Reviewers' comments:

Reviewer's Responses to Questions

**Comments to the Author**

1. If the authors have adequately addressed your comments raised in a previous round of review and you feel that this manuscript is now acceptable for publication, you may indicate that here to bypass the “Comments to the Author” section, enter your conflict of interest statement in the “Confidential to Editor” section, and submit your "Accept" recommendation.

Reviewer #1: All comments have been addressed

Reviewer #3: All comments have been addressed

2. Is the manuscript technically sound, and do the data support the conclusions?

Reviewer #1: Yes

Reviewer #3: Yes

3. Has the statistical analysis been performed appropriately and rigorously? 

Reviewer #1: Yes

Reviewer #3: Yes

4. Have the authors made all data underlying the findings in their manuscript fully available?

Reviewer #1: Yes

Reviewer #3: Yes

5. Is the manuscript presented in an intelligible fashion and written in standard English?

Reviewer #1: Yes

Reviewer #3: No

6. Review Comments to the Author

Reviewer #1: Congrats on successful revisions.

P in p values should be italicized

Check the spelling -> immunohistochemistry

Reviewer #3: The authors have done a reasonable job of responding to the earlier comments. However, this reviewer has one important comment i.e, lack of adjustment for time since diagnosis. The physical and mental health can vary substantially based on how long ago the patients were diagnosed with breast cancer (prior to their study enrollment and interview). It is recommended that the authors obtain this information and incorporate it into all their analysis.

7. PLOS authors have the option to publish the peer review history of their article (what does this mean?). If published, this will include your full peer review and any attached files.

Reviewer #1: No

Reviewer #3: No

---

## [Author Response · Author response to Decision Letter 1]

15 Apr 2024

First and foremost, we want to express our deep gratitude to the editors and reviewers who gave us valuable and practical advice. We carefully read your comments and revised our manuscript. Our responses are written in blue color. 

Thank you so much for your assistance.

..........................................................................................................................

REVIEWER REPORTS

Reviewer #1: Congrats on successful revisions.

P in p values should be italicized

Check the spelling -> immunohistochemistry

We revised these issues.

Reviewer #3: The authors have done a reasonable job of responding to the earlier comments. However, this reviewer has one important comment i.e, lack of adjustment for time since diagnosis. The physical and mental health can vary substantially based on how long ago the patients were diagnosed with breast cancer (prior to their study enrollment and interview). It is recommended that the authors obtain this information and incorporate it into all their analysis.

Regarding the time since the first diagnosis, we did collect data on this variable last year. However, when we analyzed the data, the association between patients’ utility and their time since the first diagnosis was insignificant (p-value>0.05). Therefore, when we drafted the initial manuscript, we did not report this variable.

As per the comment of Reviewer 3, we added several results involving this variable to the revised manuscript. It is noted that although in Table 4, there was a significant difference among the utility scores of patient groups (this variable was divided into four groups – a categorical variable). However, in the univariate linear regression model, when it is a numeric variable, the p-value is higher than 0.05 (coef=-0.001, p=0.059). 

As we can see from the results in Table 5, other variables, such as the stage of cancer/metastatic status, may have a stronger association with patients’ utility than the time since the first diagnosis of breast cancer. It is perfectly understandable. For example, the time since the first diagnosis of breast cancer of a patient is three months. However, this diagnosis is late and this person is a metastatic patient. 

If there is anything inappropriate or inadequate, please let us know. Thank you so much for your valuable comments.

Best regards.

---

## [Editor Report · Decision Letter 2]

18 Apr 2024

Health-state utility of patients with HER2-positive breast cancer in Vietnam: a multicenter cross-sectional study

PONE-D-23-42055R2

Dear Mr. Dai Xuan Dinh,

We’re pleased to inform you that your manuscript has been judged scientifically suitable for publication and will be formally accepted for publication once it meets all outstanding technical requirements.

Kind regards,

Ravishankar Jayadevappa, PhD, MS

Academic Editor

PLOS ONE